# Meteorological and evaluation datasets for snow modelling at ten reference sites: description of in situ and bias-corrected reanalysis data

Cecile B. Menard[1], Richard Essery[1], Alan Barr[2], Paul Bartlett[3], Jeff Derry[4] , Marie Dumont[5], Charles Fierz[6], Hyungjun Kim[7], Anna Kontu[8], Yves Lejeune[5], Danny Marks[9], Masashi Niwano[10], Mark Raleigh[11], Libo Wang[3], Nander Wever[6, 12]

[1]School of Geosciences, University of Edinburgh, Edinburgh, United Kingdom

[2] Climate Research Division, Environment and Climate Change Canada, Saskatoon, Canada; Global Institute for Water Security, University of Saskatchewan, Saskatoon, Canada

[3] Climate Research Division, Environment and Climate Change Canada, Toronto, Canada

[4] Center for Snow and Avalanche Studies, Silverton, Colorado, USA

[5] Univ. Grenoble Alpes, Université de Toulouse, Météo-France, CNRS, CNRM, Centre

d'Etudes de la Neige, Grenoble, France

[6] WSL Institute for Snow and Avalanche Research SLF, Davos, Switzerland

[7] Institute of Industrial Science, University of Tokyo, Tokyo, Japan

[8] Finnish Meteorological Institute, Space and Earth Observation Centre, Sodankylä, Finland

[9] Northwest Watershed Research Center, Agricultural Research Service, Boise, Idaho, USA

[10] Climate Research Department, Meteorological Research Institute, Tsukuba, Japan

[11] National Snow and Ice Data Center (NSIDC), University of Colorado Boulder, Boulder, Colorado, USA

[12] Department of Atmospheric and Oceanic Sciences, University of Colorado Boulder, Boulder, CO, USA





## Abstract

This paper describes in situ meteorological forcing and evaluation data, and bias-corrected reanalysis forcing data, for cold regions modelling at ten sites. The long-term datasets (one maritime, one arctic, three boreal and five mid-latitude alpine) are the reference sites chosen for evaluating models participating in the Earth System Model-Snow Model Intercomparison Project. Periods covered by the

in situ data vary between seven and twenty years of hourly meteorological data, with evaluation data (snow depth, snow water equivalent, albedo, soil temperature and surface temperature) available at varying temporal intervals. 30-year (1980-2010) time-series have been extracted from a global gridded surface meteorology dataset (Global Soil Wetness Project Phase 3) for the grid cells containing the reference sites, interpolated to one-hour timesteps and bias corrected. Although applied to all sites,

the bias corrections are particularly important for mountain sites that are hundreds of meters higher than the grid elevations; as a result, uncorrected air temperatures are too high and snowfall amounts are too low in comparison with in situ measurements. The discussion considers the importance of data sharing to the identification of errors and how the publication of these datasets contributes to good practice, consistency and reproducibility in Geosciences. Supplementary material provides

information on instrumentation, an estimate of the percentages of missing values, and gap-filling methods at each site. It is hoped that these datasets will be used as benchmarks for future model development and that their ease of use and availability will help model developers quantify model uncertainties and reduce model errors. The data are published in the repository PANGAEA and available at: https://doi.pangaea.de/10.1594/PANGAEA.897575.





## 1. Introduction

In the past decade, several long-term datasets aimed at providing high quality continuous meteorological and evaluation data for cold regions modelling have been published (Table 1). The importance of such datasets is twofold. Their primary value is scientific: they help us to understand key surface processes by enabling the development and evaluation of existing and new geophysical models for climate research and forecasting. The second, perhaps less obvious, value of having

multiple long-term datasets is for meta-research; the smaller the studies or sample size, the less likely research findings are to be true (Ioannidis, 2005). In a snow modelling context, this is corroborated by Rutter et al. (2009) who found low correlations in performance statistics for the same snow models but in different years.

   Here, we describe ten long-term datasets (Table 1) from reference sites chosen to force and to

evaluate models participating in the Earth System Model-Snow Model Intercomparison Project (ESM-SnowMIP) (Krinner et al., 2018), an international coordinated modelling effort that investigates snow schemes. ESM-SnowMIP is closely aligned with the Land Surface, Snow and Soil Moisture Model Intercomparison Project (LS3MIP; van den Hurk et al. 2016), which is a contribution to the Coupled Model Intercomparison Project Phase 6 (CMIP6) including global offline land model experiments with

meteorological forcing data provided by phase 3 of the Global Soil Wetness Project (GSWP3; Kim, 2017). Two meteorological datasets are described for each site: one compiled from on-site measurements, the other derived from GSWP3. Previous iterations of SnowMIP have provided 19 site-years of data from four sites in SnowMIP1 (Essery and Etchevers 2004) and 9 site-years of data from five sites in SnowMIP2 (Rutter et al., 2009); ESM-SnowMIP totals 136-site years of in situ data from

ten sites and 300 site-years derived from GSWP3.

   Measurement details at five of the sites have been described in dedicated publications within the last eight years. The other five sites are partially described in a number of publications which, combined, give a broad but not comprehensive overview of the data. All of the in situ measurements and the GSWP3 data are freely available either on the web or on request but, previously, post-processing

would have been required to homogenize the in situ datasets compiled by different teams or to downscale the reanalyses. This situation causes two major issues. Firstly, different modelling teams are likely to apply different post-processing methods, leading to numerous versions of the same dataset being used for scientific studies. Secondly, although time spent identifying and processing data has never been quantified in scientific literature to our knowledge, it is a well-known but under-

acknowledged time consuming task for modelers.



The aim of this collaborative work is to provide easy-to-use, quality-controlled data in a format adopted by the climate modelling community to facilitate consistency, continuity and reproducibility in snow research (Menard and Essery, 2019). As such, it complies with efforts in geosciences to foster best practices on data accessibility and documentation (Gil et al., 2016). The seven teams who collated

the in situ datasets have provided updates since previous publications and details about instrumentation, gaps in the original data and methods for gap filling. Such additions are first steps towards being able to quantify uncertainty in observed data, without which "meaningful evaluation of a model is impossible" (Clark et al., 2011). Similarities and differences between sites are discussed in individual measurement sections.

**Table 1: Data ownership and reference papers for the sites. Asterisks denote dedicated data description papers; the others are modelling papers in which a short description of a site is included.**

2. Data

Of the ten sites, two are in the European Alps (Col de Porte and Weissfluhjoch), three are in the mountains of the Western USA (Reynolds Mountain East, Senator Beck and Swamp Angel), three are in the Canadian boreal forest (the Boreal Ecosystem Research and Monitoring Sites, BERMS, the acronym hereafter collectively describing the Old Aspen, Old Black Spruce and Old Jack Pine sites), one is sub-arctic (Sodankylä) and one is urban (Sapporo). Broad geographic characteristics of the sites

are described in Table 2. The climate of each site is described by a snow cover classification and the Köppen climate classification, based on seasonal precipitation and air temperature. Information about data ownership and references are given in Table 1. Most sites are in artificial forest gaps or in sheltered environments. While this facilitates measurements of precipitation prone to large errors in windy environments, implications for the other meteorological variables are discussed in their

dedicated sections.

Three of the sites (SAP, SOD and WFJ) are located near staffed research stations, which allows frequent (daily to sub-weekly) and regular maintenance of the instruments. Col de Porte, Reynolds Mountain East and the sites in the Senator Beck basin (SWA and SNB) are accessible from nearby research facilities allowing regular (weekly to fortnightly) maintenance visits. Intensive monitoring associated

with the BERMS project took place in the first years after the instruments were installed, but visits to the sites during winter have become sporadic.



Both meteorological and evaluation data contain uncertainties and errors, partly due to instrument accuracy and calibration, gap-filling of missing data or subjective choices; precipitation is a notable example (See Section 2.1.3 for details). Fully quantifying these uncertainties and errors is beyond the scope of this paper, but the comprehensive list of instruments and information about missing data and gap-filling available in the supplementary material provides data users with an indication of weaknesses in the data at each site.

**Table 2: Geographic characteristics of the ten sites.**

### 2.1 Meteorological forcing data

All of the models participating in ESM-SnowMIP (Krinner et al. 2018) operate on energy balance principles, requiring incoming shortwave and longwave radiation fluxes, solid and liquid precipitation rates, air temperature, humidity, wind speed and air pressure as forcing data. Pressure is used by models to calculate air density and vapour pressure, but the temporal coefficient of variation in pressure is always very small; averages for the site elevations can be used where continuous measurements are not available (CDP, RME, SNB and SWA). At Col de Porte and Sapporo, where data outside of the snow season have not been published, all meteorological data are filled with downscaled (CDP) and bias-corrected (SAP) meteorological reanalysis data (publication of summer data for Col de Porte started in 2015; Lejeune et al., 2018). Figure 1 shows monthly averages of all meteorological forcing variables except air pressure at all sites. Details of each variable are given in the following sections.

**Figure 1: Climatological monthly averaged meteorological forcing data. Wind speeds at all sites are normalised at 10 m height.**

#### 2.1.1    Air temperature

The range of air temperature at all sites is shown in

Figure *2*. Sapporo has the highest annual mean (9.3°C) and minimum (-15.8°C) temperatures, although Col de Porte is generally warmer from December to February. The lowest and only annual mean temperature below freezing (-1°C) is at Senator Beck. The coldest winters are at BERMS (the lowest



temperature recorded, -41°C, is at Old Black Spruce) and Sodankylä, where most years see temperatures down to -35°C.

In terms of instrumentation, Col de Porte is the only site at which the temperature sensor is moved (approximately weekly) to keep it at a constant height above the snow; otherwise, it is recommended that measurement heights use in models should be adjusted according to observed or simulated snow depths because this can have a significant impact on turbulent flux computations. Depending on wind speed and solar radiation, unventilated instruments can overestimate air temperature by up to a daytime average of 2.5°C (Georges and Kaser, 2002) or up to 10°C for individual measurements (Huwald et al., 2007). Such errors are not corrected for in Reynolds Mountain East, the Senator Beck basin sites or Sodankylä (temperature sensors at the other sites are artificially ventilated).

**Figure 2: Boxplots of air temperature including means (red dashed line) at all sites. Outliers beyond 1.5 times the interquartile range (25th to 75th percentiles) are marked with circles.**

### 2.1.2    Incoming shortwave and longwave radiation

Sodankylä is the only site situated above the Arctic Circle and therefore has continuous periods without incoming solar radiation in winter (14 days) and uninterrupted daylight in Spring/Summer (44 days). Longwave radiation depends, amongst others, on air temperature, water vapour, cloud cover and altitude. In general, the order of incoming longwave radiation ($LW$) between sites is very close to that of air temperature ($T_a$). However, other influences can be seen in the Senator Beck Basin sites, which have lower $LW$ but higher $T_a$ than BERMS because the former are drier. The same reversal occurs between Sodankylä and Reynolds Mountain East.

With the exception of precipitation, radiation measurements are the most prone to errors and/or missing data because snow can settle on upward-looking sensors. In the absence of wind to displace the snow or if the instruments do not have a heating and ventilation system to prevent snow accumulation, data are only reliable after the instruments have been wiped clean. Maintenance frequencies are described in the introduction to Section 2. Methods for gap-filling during snowfall events or while instruments are obstructed by snow vary; details for all sites are in the supplementary material.

Radiation in Sodankylä is measured above the canopy, but evaluation data are measured in a nearby clearing. For consistency, shortwave radiation was modified to account for the effects of shortwave shading and longwave emission from nearby trees (Essery et al. 2016). At Reynolds Mountain East, longwave radiation measurements started in 2002. For consistency across the RME dataset, which





starts in 1988, all longwave radiation is modelled but measured data are used to provide information
on seasonal and diurnal variations (e.g. cloud cover, turbidity, canopy and terrain exposure
conditions). Details of the methods used to model $LW$ are in Reba et al. (2011).

### 2.1.3    Precipitation

Snowfall measurements are often underestimates and prone to large errors because much is lost to
sublimation or displaced by wind. Such difficulties are acknowledged by the World Meteorological
Organization (WMO) which, rather than imposing a standardized method, advises that adjustment
methods be chosen depending on environmental conditions and gauge types (Goodison et al., 1998;
Nitu et al., 2018). As detailed in the supplementary material, precipitation at all sites is measured
either with tipping buckets or weighing gauges and six different methods are applied by the seven
collecting teams to correct for undercatch: yearly or constant scaling factors, model simulations,
matching against SWE or replicate gauges. Furthermore, as weighing gauges do not provide
information on the type of precipitation, choices also have to be made about how to partition snowfall
and rainfall. Figure 3 shows how the different methods used at each site affect the solid fraction of
precipitation as a function of air temperature; total precipitation at Swamp Angel and Senator Beck
are assumed to be same because of their proximity so only the latter only is shown. Partitioning
methods include using dew point (RME, SAP, SWA, SNB) or air temperature (BERMS, SOD, WFJ)
functions or thresholds, and ancillary data such as snow depth and albedo measurements (CDP); more
information is provided in the supplementary material.

**Figure 3: Fraction of precipitation falling as snow at different temperatures, as imposed on the in
situ data and fitted to the GSWP3 data.**

Frequent summer snowfall at Weissfluhjoch and early autumn snowfall at Col de Porte can form snow
cover that melts before the winter snow pack accumulates, thus causing discrepancies between
annual snowfall and peak SWE. At BERMS, such discrepancies are mostly accounted for by snowfall
intercepted by the canopy that is lost through sublimation.

### 2.1.4    Wind speed

Wind speeds provided in the datasets are measured at variable heights but were normalised to 10 m
height assuming a logarithmic wind profile for Fig. 1-g such that



$$u(z_2) = u(z_1)\frac{ln((z_2 - d)/z_0)}{ln((z_1 - d)/z_0)}$$

where $u$ is wind speed at measured ($z_1$) and normalised ($z_2$) heights, $d$ is a displacement height (2/3 of vegetation height at BERMS, 0 at other sites) and $z_0$ is a roughness length (1/10 of vegetation height at BERMS and 0.1 m at the other sites). The three sites with the lowest wind speed in Fig. 1-g are situated in forest gaps. There is little difference between mean wind speeds at Sodankylä and Col de

Porte. Wind speed at Sodankylä is measured above the canopy but was scaled down to 2 m height against an anemometer installed for one week in the ~2400 m² clearing. The approximately 60 x 50 m dedicated experimental area at Col de Porte is situated in the southeast corner of a larger clearing (270 x 360 m) within a spruce forest. As mentioned in Morin et al. (2012), all trees sheltering the north side of the experimental area were cut in summer 1999; mean wind speed at 10 m height was 1 m s⁻¹

prior to the event but 1.26 m s⁻¹ afterwards. Nevertheless, a Mann-Kendall (MK) test shows a significant increasing trend in wind speed from 1999 onwards despite the tree growth mentioned in Lejeune et al. (2018). Average wind speed at the exposed Senator Beck is the highest of all the sites and is almost four times more than at the nearby sheltered Swamp Angel. MK shows a significant decreasing trend in wind speed at Reynolds Mountain East and increasing trends at the Senator Beck

basin sites. At Reynolds Mountain East and Weissfluhjoch, but more prominently at Sapporo and Senator Beck, wind speed is higher in winter; it is highest in spring at the BERMS Old Aspen and Old Black Spruce. Col de Porte, Swamp Angel and Sodankylä do not show strong seasonal variability.

### 2.1.5    Relative Humidity


Humidity is measured at all of the sites except Weissfluhjoch using capacitive sensors. These sensors respond to changes in relative humidity (Anderson 1995), but vapour fluxes in models are driven by specific humidity gradients. Conversion is therefore required from relative to specific humidity. At temperatures below 0°C, there are two possible definitions of relative humidity because of the

different saturation vapour pressures over water and ice. Sensors calibrated following the WMO convention of reporting relative humidity with respect to water at all temperatures are used at most of the sites. The consequences of this choice (shown in Fig. 4) are not very significant at warmer sites such as Col de Porte or Sapporo but are clear in data from colder sites such as Old Jack Pine and Swamp Angel; relative humidity is never observed much above the ice saturation point for a particular

temperature. A chilled mirror dew point hygrometer is used at Weissfluhjoch, and reported relative humidity can reach 100% or higher even at sub-zero temperatures. In homogenizing the datasets,





relative humidity has been limited to a maximum of 100% and converted to specific humidity using the site calibrations.

**Figure 4: Example scatter plots of relative humidity against temperature for four of the sites. The**
**solid lines show ice saturation at temperatures below 0°C and water saturation above. Lines of**
**constant specific humidity near the upper end of the data ranges are dashed.**

### 2.2 Evaluation data

#### 2.2.1      Snow depth and water equivalent


**Figure 5: Monthly climatological averages of manual snow water equivalent measurements.**

**Figure 6: Daily climatological averages of snow depth measurements at all sites.**

Although automatic sensors are increasingly being used to measure SWE, the most reliable methods to obtain snow mass are still manual (Pirazzini et al., 2018). They work by weighing snow mass in
samplers of known volume or area, such as small cutters in snow pits or tubes to extract vertical snow cores. Nevertheless, such measurements are prone to errors: wet snow can stick to instruments, manual measurements can never be replicated in the same place because they are destructive, and subjectivity and skill do play a part so consistency can be hard to achieve if multiple people collect the data.

Replicate measurements of SWE and snow depth can be used to estimate uncertainty, which can be caused by measurement errors, spatial variability or a combination of both. At Col de Porte, three replicate weekly snow pits are available, two of which are used to calibrate automatic SWE measurements. Their mean standard deviation is 17 kg m$^{-2}$; although standard deviation increases with increasing snow amount, it is generally less than ten percent of mean SWE. At Reynolds Mountain
East, there is a snow pillow next to a snow course that is visited approximately 10 to 15 times during the snow season. Root mean square difference (RMSD) between the two methods is 40 kg m$^{-2}$, for annual maximum SWE ranging from 186 kg m$^{-2}$ (1992) to 838 kg m$^{-2}$ (1989). RMSD in snow depth can be calculated at all sites as all have both automatic and manual measurements. Results are shown in Table , along with maximum and minimum peak snow depth to normalise the difference. At Senator





Beck, the snow pits cannot be collocated with the automated snow depth so the spatial variability of
snow is intrinsic to any comparisons between the manual and automated measurements.

**Table 3: Root mean square difference between manual and automatic snow depth measurements,
maximum yearly snow depth and minimum yearly snow depth for all sites.**

Climatological averages of measured SWE and snow depth are shown in Figs. 5 and 6 respectively.
Although all sites are situated in the Northern Hemisphere and only one is above the Arctic Circle, the
snow season characteristics provide a diverse range of scenarios for the evaluation and development
of snow models e.g. cold sites (e.g. SNB, SWA, SOD) with a well-defined snow season (snowpack
building in autumn and winter, melting in spring/summer), warmer sites with occasional early- to mid-
season snowmelt (CDP and SAP), and forest sites with interception of snowfall by the canopy (BERMS).


### 2.2.2    Albedo

**Figure 7: Daily averaged albedo over time (a) and as a function of snow depth (b) at all sites except
RME and SOD.**

Reflected shortwave radiation is measured at all sites except Sodankylä and Reynolds Mountain East,
thus allowing calculations of albedo. Daily effective albedos (Fig. 7-a) have been calculated at all sites
with reflected shortwave radiation measurements by the method used by Morin et al. (2012) for Col
de Porte. Hourly data are rejected during snowfall, if incoming shortwave radiation is less than 20 W
$m^{-2}$ or if reflected shortwave radiation is less than 2 W $m^{-2}$. For days with more than five hours of data
remaining after rejection, an albedo is calculated by dividing the sum of reflected shortwave radiation
measurements by the sum of incoming shortwave radiation measurements.

Peak snow depth and highest albedo do not coincide at BERMS in Fig. 7-b because the highest albedo
occurs when the most snow is intercepted by the canopy during the coldest month of the year
(January), not at the end of the snow accumulation period. Curves of albedo against snow depth show
hysteresis at all of the sites, with snow cover of the same depth having lower albedo when melting
than when accumulating. Albedos of melting snow can be much lower at Senator Beck and Swamp
Angel than at other non-forested sites with comparable snow depths because of frequent dust storms
dirtying the snow surface (Painter et al., 2012). Although it is not obvious from Fig. 7 in the absence
of nearby sites without snow impurities, high concentrations of black carbon are found in the Sapporo
snowpack (Aoki et al., 2011). Model simulations suggest that impurities reduce albedo at Sapporo by



0.05 in winter and 0.18 during melt (Niwano et al., 2012). Information about errors and uncertainties in albedo due to incoming radiation measurements is in Section 2.1.2.

### 2.2.3  Surface and Soil temperature


**Figure 8: Daily climatological averages of surface temperature (a), soil temperature (b), and differences between air and soil temperatures (c). Soil temperatures are shown at 30 cm depth at RME and at 10 cm depth at all other sites.**

Successive IPCC reports have noted that Earth System Models often underestimate soil temperatures
at high latitudes (Randall et al., 2007; Flato et al., 2013; Koven et al., 2013). This has implications for assessing the permafrost carbon feedback, i.e. the amplification of surface warming from carbon emissions released by thawing permafrost. Long term datasets are therefore essential to evaluate model performance and to improve model representations of soil / atmosphere interactions.

Surface temperature (Fig. 8-a) and soil temperatures (Fig. 8-b) are available at eight of the sites.
Surface temperature was calculated from measured outgoing longwave radiation assuming blackbody radiation except at the Senator Beck basin sites where infrared temperature sensors measurements are used. Unlike at Col de Porte, Sapporo and Weissfluhjoch where the pyranometers measuring outgoing longwave radiation are above snow cover, the instruments at BERMS are above the canopy.

The strong insulating effect of snow is apparent in Fig. 8 b-c for all sites with average winter air
temperatures below 0°C. Even at a shallow depth (10 cm), daily averaged winter soil temperatures remain above freezing at all sites except Old Jack Pine and Sodankylä, although the soil does freeze in some individual years. They are also the only two sites not to have soil temperatures at 10 cm depth (or 20 cm, not shown) plateau during the snow season. They do show the highest annual ranges of temperatures, with climatologically averaged winter temperatures down to -5°C and summer
temperatures above 12°C.

### 3.  Large-scale meteorological forcing data for reference site simulations

The Land Surface, Snow and Soil Moisture Model Intercomparison Project (LS3MIP; van den Hurk et
al. 2016) contribution to CMIP6 includes global offline land model experiments with meteorological forcing data provided by phase 3 of the Global Soil Wetness Project (GSWP3; Kim, 2017). GSWP3

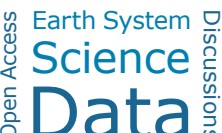

forcing data were generated by a run of the Global Spectral Model at T248 (approximately 50 km) resolution nudged at each pressure level with meridional- and zonal-wind and air temperature from the 20[th] Century Reanalysis (Compo et al. 2011), followed by bias corrections using observations. All of the variables required for forcing land surface models are provided on a 0.5° global grid and three-hour timesteps.

For ESM-SnowMIP, 1980-2010 forcing data have been extracted for GSWP3 grid cells containing reference sites and interpolated to one-hour timesteps. The longer time period provides more variability for investigating the sensitivity of models to trends in forcing data. These data would also allow rerunning LS3MIP experiments at reference sites with models that do not have capabilities for global runs, but a complication is immediately apparent from the comparisons of site and grid data in Fig. 9. The maritime, boreal and Arctic sites (SAP, OAS, OBS, OJP, SOD) are in areas with low relief and lie close to the mean elevations of their GSWP3 grid cells, but snow study sites in mid-latitude mountains (CDP, RME, SNB, SWA, WFJ) are typically established at higher elevations with longer snow seasons; most of the ESM-SnowMIP mountain sites are hundreds of metres higher than grid elevations (Fig. 9-a). Consequently, GSWP3 temperatures at the mountain sites are too high (Fig. 9-b), total precipitation is too low (Fig. 9-c), snowfall is much too low (Fig. 9-d), and bias corrections are required for model forcing.

**Figure 9: Comparisons between elevations (a), temperatures (b), total precipitation (c) and snowfall (d) at ESM-SnowMIP reference sites and corresponding GSWP3 grid cells. Triangles identify mountain sites.**

Bias corrections have been applied to all GSWP3 meteorological variables at all sites. Quantile mapping was used to correct relative humidity within the 0-100% range, but only mean biases for overlapping data periods were removed from the other variables to retain the interannual and shorter variability in the large-scale forcing; the aim is to stay as close as possible to the global GSWP3 simulations without introducing gross elevation-dependent errors in site simulations. Offsets were applied to air temperature, pressure and longwave radiation data, and multipliers were applied to precipitation, wind speed and shortwave radiation data to avoid negative or spurious non-zero values. Site wind speeds were first normalized to the GSWP3 10 m reference height using a logarithmic profile and an assumed 0.1 cm roughness length.

Total precipitation rate $P_r$ in each timestep was repartitioned into snowfall rate $S_f = f_s P_r$ and rainfall rate $R_f = (1 - f_s)P_r$ depending on corrected air temperature $T$ using a logistic curve



$$f_s = \frac{1}{1 + \exp[(T - T_0)/T_1]}$$

with site-dependent parameters $T_0$ and $T_1$ fitted to unadjusted GSWP3 data (Table 4). Figure 3 shows that the logistic curve fits the GSWP3 data well at all sites with the exception of Sapporo, which has the unusual feature of some precipitation at low temperatures falling as rain. Inspection of the gridded GSWP3 data shows occurrences of freezing rain in coastal areas.

**Table 4: Precipitation phase factors fitted to GSWP3 data at site locations (Senator Beck and Swamp**
**Angel are located within the same 0.5° grid cell).**

Annual mean temperature and snowfall variations are shown in Figs. 10 and 11 for the in situ and bias-corrected GSWP3 data at all sites. Although only mean errors for the periods of overlap have been removed, there is generally good correlation between annual means of GSWP3 data and site
observations for overlapping years. Table  gives linear trends fitted to the in situ and bias-corrected GSWP3 annual mean temperatures and snowfall. 1998-2009 observations at BERMS show decreasing temperatures and increasing snowfall after the Saskatchewan drought of the early 2000s, but there are negligible trends in the longer GSWP3 series. Sapporo also has increasing snowfall in recent years but little trend in GSWP3. Some sites show stronger warming trends in the GSWP3 data, which will be
useful for investigating modelled snow responses to warming.

**Table 5: Trends in annual mean temperatures and snowfall from in situ and GSWP3 data. Bold trends are statistically significant (Mann-Kendal p< 0.05).**

**Figure 10: Annual mean temperatures and fitted trends for years starting on 1 October at reference sites from GSWP3 and in situ data. Numbers show correlation ($r$) between GSWP3 and in situ air**
**temperature for the $n$ complete years of overlap.**

**Figure 11: Annual snowfall and fitted trends for years starting on 1 October at reference sites from GSWP3 and in situ data.**

## 4. Discussion

A number of errors were identified in the datasets in the course of the study. Firstly, we noted that snowfall at the Old Aspen was much lower than at the Old Black Spruce and Old Jack Pine during the 2007 / 2008 winter. It was subsequently found that a gauge malfunction in November and December 2007 was not identified at the quality control (QC) stage. Secondly, two other errors were identified



by decomposing time series: trend analyses showed an increase in wind speed at the Senator Beck

basin sites from October 2012 to the end of the dataset in October 2015. Both sites measure wind

speed at two heights; the lower wind speed measurements were used for the first seventeen years of

the dataset, but the upper wind speed was accidentally used for the last three years. At the same sites,

instrument re-calibration led to a small but statistically significant increasing trend in longwave

radiation. These errors were included in the preliminary ESM-SnowMIP results shown in Krinner et al.

(2018); erroneous years will either be neglected in future publications or models will be forced with

the corrected datasets which are published alongside this paper (see Section 6).

While unfortunate, such errors are symptomatic of long-term data sets for which consistent

maintenance and data collection is problematic. Firstly, by definition, long-term monitoring stations

might have been installed before metadata were kept electronically (and before the word "metadata"

was invented in 1983; Merriam-Webster, 2018) and when information about changes of instruments

or re-calibrations were in notebooks which might never have been digitised, have now been lost or

never even existed. Equally, improvements in data storage capacities mean that temporal sampling

intervals are shorter than they were. For example, measurements at Reynolds Mountain East were

initially made every 15 minutes and averaged to hourly values; currently, 10 second samples and 5

minutes averages are aggregated to hourly values for most variables. Such factors are known to affect

the values of meteorological variables (Hupet and Vanclooster, 2001) but it is beyond the scope of this

study to attempt to quantify their contributions to errors or variations in the datasets. Secondly,

immediate use of the data allows instrument malfunctions to be identified quickly. For example, a

power supply failure was not identified at Sodankylä for 52 days in September and October 2011

because data were being collected but not used; more frequent QC checks are now in place. Thirdly,

long-term monitoring stations are susceptible to funding cycles and to changes in climate change

policies by successive governments. For example, the BERMS sites, which were established in 1994,

had the most frequent site visits from 2001 to 2008, but changing priorities led to less frequent snow

surveys after 2008 with only one in the 2009/2010 snow season. Finally, while automated QC

protocols are in place, some checks require a subjective interpretation of the data and can therefore

depend on just one person to identify errors due to malfunction, snow deposition on instruments etc.

Reliance on subjectivity or local knowledge – which in some cases is advocated as mentioned in

Section 2.1.3 to choose the best method to correct undercatch in precipitation – diminishes the

likelihood of the dataset being reproducible. In a discipline like geoscience where uncertainties and

errors are required to be quantifiable, it is important to acknowledge that subjectivity is not. The

closest estimate comes from a survey in which more than 40% of scientists in the field of Earth and

Environment admitted to failing to have reproduced their own experiments (Baker, 2016); the figure



increased to more than 60% when trying to reproduce other researchers' experiments. Nevertheless, human errors, or more appropriately "mistakes", are not exclusive to data processing: Menard et al.
(2015) identified mistakes in the description files of the land surface model JULES that caused it to underperform considerably.

A recent and growing push towards standardising methods for data sharing and publishing may lead to errors being identified more systematically as more people have access to data. One of the advantages of open source software is that bugs are reported by users and their correction is, at times,
a community effort which allows software to be improved quickly (Wu et al., 2016). Sharing of geoscientific models' source code, although still a fairly recent development compared to the field of engineering software, has equally led to model improvements through the identification and fixing of bugs beyond the model development teams (David et al., 2016; Samuel Morin, personal communication about the Crocus snow model). One might expect a similar trend for data sharing
where identifying errors becomes an asset to the community because, as mentioned by Gil et al. (2016) in their proposal for a framework for best practices in the publication of data papers, "data sharing makes authors double-check their work, improving science at the first stage as well as future reuse". The more data are used, the more likely it is that mistakes, errors and uncertainties are identified, and the less likely it will be that model results can, according to Clark et al. (2011) "at best
be merely attributed to a nebulous mix of data and structural errors"; to this we can also add human errors.

## 5. Conclusion

It is hoped that one of the legacies of ESM-SnowMIP will be for the datasets presented in this paper
to be used as benchmarks for model development and to facilitate improvements in snow modelling. Cold region processes have been a major source of uncertainties in previous IPCC reports. The sparsity of long-term high quality datasets in cold regions in the past may have contributed to this if one considers that ESMs are run globally but their snow schemes are generally evaluated at a small number of sites; the first iteration of SnowMIP (Etchevers et al., 2002) sixteen years ago included only
one long-term (15-year) dataset and three short-term (less than two snow seasons) ones. Meta-research argues that it is misleading to emphasize statistically significant findings of any single team; what matters instead is the totality of the evidence (Ioannidis, 2005). It is equally misleading to draw conclusions on model performance when models are evaluated only at one or two sites for one or two years. The ease-of-use and availability of the datasets presented here, as well as further ESM-





SnowMIP reference sites which will be located in more challenging conditions, should help model
developers quantify – and reduce – model uncertainties and errors.

## 6. Data availability and archiving

The data presented and described in this paper are available in the data repository PANGAEA:

https://doi.pangaea.de/10.1594/PANGAEA.897575

## Author Contribution

CM led the writing of the paper and archived the data. RE prepared the data in the standardized format
described in Section 6, bias-corrected GSWP3 data and wrote Section 3. The supplementary material
(metadata) and data provision are attributable to: MD and YL for CDP, AB and PB for BERMS, DM for
RME, MN for SAP, JD and MR for SNB and SWA, AK and RE for SOD and CF for WFJ. HK provided GSWP3
data, which were extracted for sites and interpolated by LW. All co-authors provided comments which
contributed to the paper.

## Acknowledgments

The authors would like to thank all the staff and students who have collected the data presented in
this paper over the years. Work by CM and RE was supported by NERC grant NE/P011926/1.
CNRM/CEN is part of Labex OSUG@2020 (ANR10 LABX56). CDP is part of Observatoire des Sciences
de l'Univers de Grenoble (OSUG), Observation pour l'Experimentation et la Recherche en
Environnement CryObsClim and Systemes d'Observation et d'Experimentation au long terme pour la
Recherche en Environnement des glaciers, GlacioClim. CDP contributes to OZCAR (Observatoires de la
Zone Critique Applications et Recherches), one of the French components of the eLTER European
Research Infrastructure (International Long-term Ecological Research Networks). It is also labeled as a
member of the World Meteorological Observation Global Cryospheric Watch Cryonet network and of
the INARCH network. H. Kim acknowledges support by Grant-in-Aid for Specially promoted Research
16H06291 from JSPS. M. Niwano was supported in part by (1) the Japan Society for the Promotion of
Science through Grants-in-Aid for Scientific Research numbers JP16H01772 (SIGMA project),
JP15H01733 (SACURA project), JP17K12817, JP17KK0017, JP18H03363, and JP18H05054; (2) the
Ministry of the Environment of Japan through the Experimental Research Fund for Global
Environmental Research Coordination System; (3) the Institute of Low Temperature Science, Hokkaido



University, through the Grant for Joint Research Program (18S007 and 18G035). The sites at the Senator Beck Basin are maintained by the Center for Snow and Avalanche Studies with development funding from the U.S. National Science Foundation (ATM-0431955) and the USDA-Forest Service. A. Barr and P. Bartlett acknowledge financial support from the Climate Research Division of Environment

and Climate Change Canada, and field and data management support from Joe Eley, Charmaine Hrynkiw, Dell Bayne, Natasha Neumann, Erin Thompson and Steve Enns. Sodankylä is a member of WMO Global Cryosphere Watch Cryonet network.

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



Tables

| Site | Short name | Data provider | Reference paper |
|---|---|---|---|
| **Col de Porte France** | CDP | Météo-France, France | Morin et al. (2012)* Lejeune et al. (2018)* |
| **Old Aspen, Old Black Spruce Old Jack Pine Saskatchewan, Canada** | OAS OBS OJP (BERMS collectively) | Environment and Climate Change Canada, Canada | Bartlett et al. (2006) |
| **Reynolds Mountain East Idaho, USA** | RME | USDA Agricultural Research Service, USA | Reba et al. (2011)* |
| **Sapporo Japan** | SAP | Meteorological Research Institute, Japan Meteorological Agency, Japan | Niwano et al. (2012) |
| **Senator Beck Swamp Angel Colorado, USA** | SNB SWA (Senator Beck basin collectively) | Center for Snow and Avalanche Studies, USA | Landry et al. (2014)* |
| **Sodankylä Finland** | SOD | Finnish Meteorological Institute, Finland | Essery et al. (2015)* |
| **Weissfluhjoch Switzerland** | WFJ | WSL Institute for Snow and Avalanche Research, Switzerland | Wever et al. (2015) |

*Table 1: Data ownership and reference papers for the sites. Asterisks denote dedicated data description papers; the others are modelling papers in which a short description of a site is included.*







| Short name | Latitude (°N) | Elevation (m) | Vegetation type | Soil type | Snow cover classification | Köppen climate classification |
|---|---|---|---|---|---|---|
| **CDP** | 45.30 | 1325 | Grassy meadow surrounded by coniferous forest | Sandy clay loam | Alpine | Warm-summer humid continental climate |
| **OAS** | 53.63 | 600 | 21 m high aspen forest. Thick understory of 2 m high hazelnut. | 10 cm organic litter and peat over sandy clay loam | Taiga | Warm-summer humid continental climate |
| **OBS** | 53.99 | 629 | 12 m high black spruce forest. Sparse understorey. | Peat over sand and sandy loam | Taiga | Warm-summer humid continental climate |
| **OJP** | 53.92 | 579 | 14 m high forest. Sparse understorey. | Sand | Taiga | Warm-summer humid continental climate |
| **RME** | 43.19 | 2060 | Clearing (short grass) in an alpen/fir grove | Silty clay | Alpine | Warm-summer humid continental climate |
| **SAP** | 43.08 | 15 | Short grass | Clay | Maritime | Hot summer continental climates |
| **SNB** | 37.91 | 3714 | Alpine tundra | Thin soil and exposed bedrock | Alpine | Polar and alpine (montane) climates |
| **SOD** | 67.37 | 179 | Clearing (short heather and lichen) in coniferous forest | Sand | Taiga | Subarctic climate |
| **SWA** | 37.91 | 3371 | Clearing (short grass) in subalpine forest | Colluvium | Alpine | Subarctic climate |
| **WFJ** | 46.83 | 2536 | Barren | Moraine | Alpine | Polar and alpine (montane) climates |

*Table 2: Geographic characteristics of the ten sites.*




| Sites | RMSD (m) | Max peak yearly Snow depth (m) | | Min peak yearly snow depth (m) | |
|---|---|---|---|---|---|
| | | Manual | Automatic | Manual | Automatic |
| CDP | 0.11 | 2.09 | 2.03 | 0.53 | 0.60 |
| OAS | 0.06 | 0.60 | 0.68 | 0.32 | 0.34 |
| OBS | 0.05 | 0.61 | 0.55 | 0.29 | 0.30 |
| OJP | 0.06 | 0.54 | 0.61 | 0.25 | 0.31 |
| RME | 0.08 | 2.02 | 2.14 | 1.06 | 1.02 |
| SAP | 0.08 | 1.22 | 1.20 | 0.62 | 0.52 |
| SNB | 0.27 | 2.37 | 2.30 | 1.52 | 1.37 |
| SOD | 0.04 | 1.03 | 1.02 | 0.65 | 0.61 |
| SWA | 0.11 | 2.66 | 2.90 | 1.66 | 1.78 |
| WFJ | 0.05 | 3.56 | 2.95 | 1.82 | 1.82 |


*Table 3: Root mean square difference between manual and automatic snow depth measurements, maximum yearly snow depth and minimum yearly snow depth for all sites.*

| Site | $T_0$ (°C) | $T_1$ (°C) |
|---|---|---|
| CDP | 3.08 | 1.13 |
| OAS | -1.73 | 1.63 |
| OBS | -1.14 | 1.81 |
| OJP | -1.32 | 1.76 |
| RME | -2.00 | 1.48 |
| SAP | 3.72 | 1.48 |
| SNB / SWA | -3.01 | 2.05 |
| SOD | 2.52 | 1.16 |
| WFJ | 0.39 | 1.47 |

*Table 4: Precipitation phase factors fitted to GSWP3 data at site locations (Senator Beck and Swamp*
*Angel are located within the same 0.5° grid cell).*

| | Temperature trend (°C/year) | | Snowfall trend (%/year) | |
|---|---|---|---|---|
| Site | In situ | GSWP3 | In situ | GSWP3 |
| CDP | 0.01 | **0.04** | -0.56 | -1.25 |
| OAS | -0.11 | 0.01 | 1.54 | -0.02 |
| OBS | -0.16 | 0.01 | 3.98 | -0.07 |
| OJP | -0.15 | 0.01 | 2.02 | -0.07 |
| RME | 0.02 | **0.06** | 0.31 | -1.42 |
| SAP | 0.02 | **0.04** | **5.01** | -0.06 |
| SNB | 0.05 | 0.01 | -2.10 | -0.55 |
| SOD | 0.08 | **0.07** | 0.05 | -0.29 |
| SWA | 0.05 | 0.01 | -1.56 | -0.53 |
| WFJ | 0.03 | **0.03** | -1.47 | **-0.88** |

*Table 5: Trends in annual mean temperatures and snowfall from in situ and GSWP3 data. Bold trends are statistically significant (Mann-Kendal $p < 0.05$).*



Figures

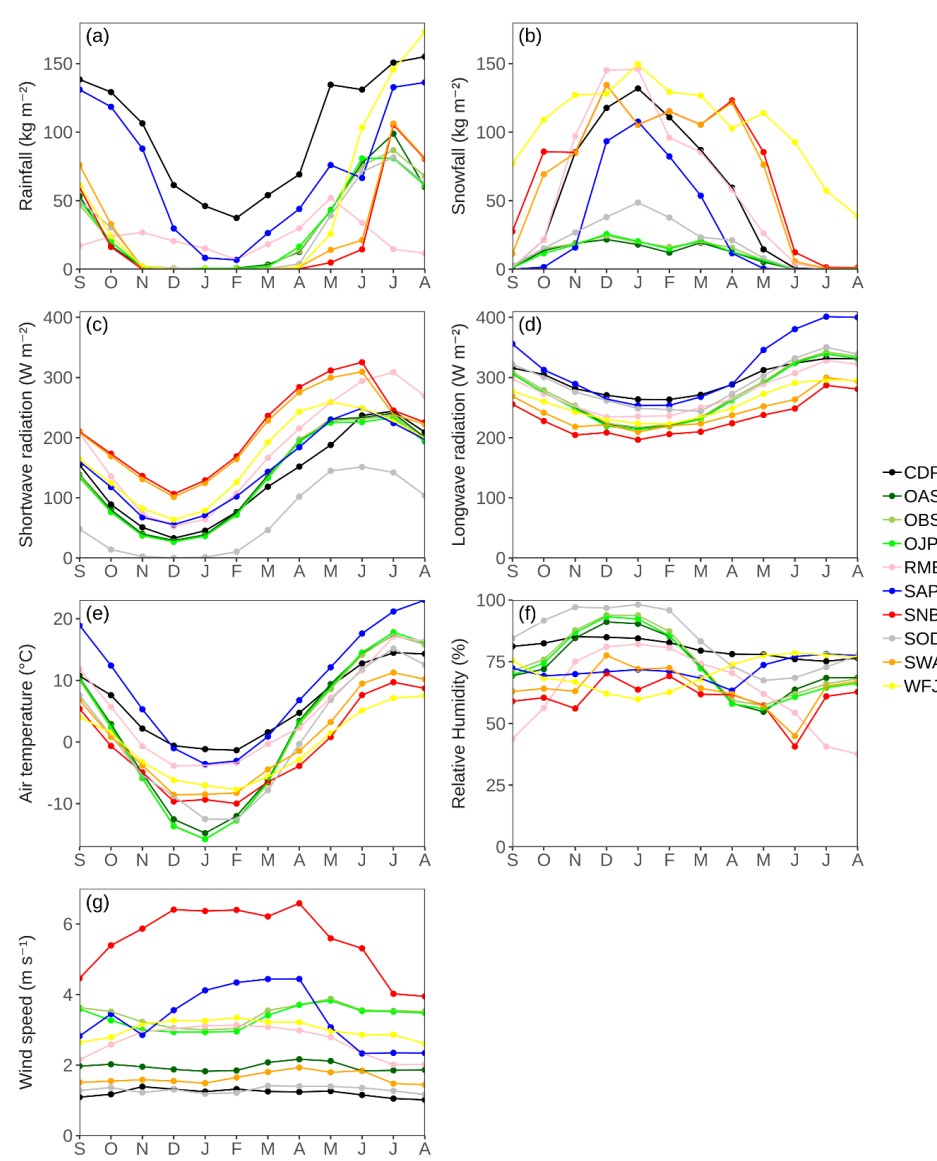

Figure 1: Climatological monthly averaged meteorological forcing data. Wind speeds at all sites are normalised at 10 m height.





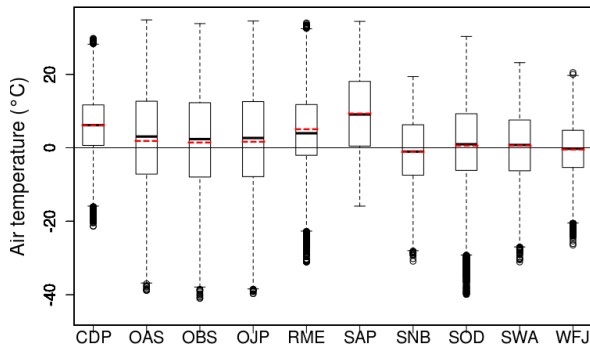

*Figure 2: Boxplots of air temperature including means (red dashed line) at all sites. Outliers beyond 1.5*

*times the interquartile range (25$^{th}$ to 75$^{th}$ percentiles) are marked with circles.*





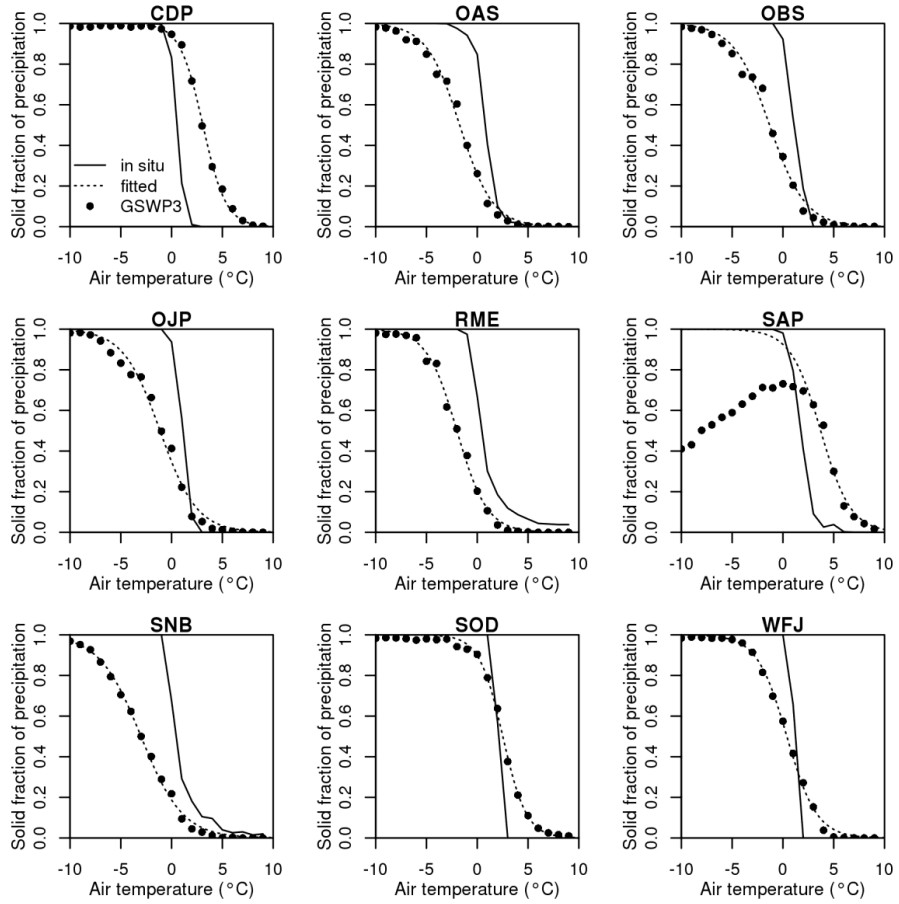

*Figure 3: Fraction of precipitation falling as snow at different temperatures, as imposed on the in situ data and fitted to the GSWP3 data.*




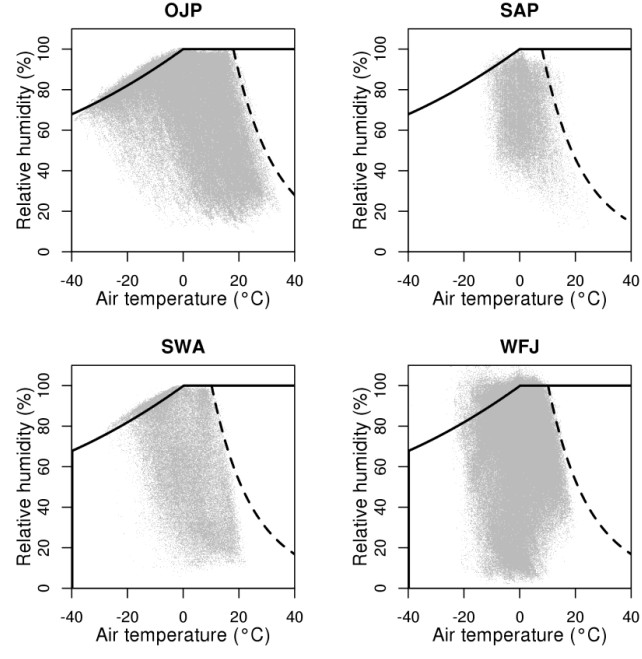


Figure 4: Example scatter plots of relative humidity against temperature for four of the sites. The solid lines show ice saturation at temperatures below 0°C and water saturation above. Lines of constant specific humidity near the upper end of the data ranges are dashed.



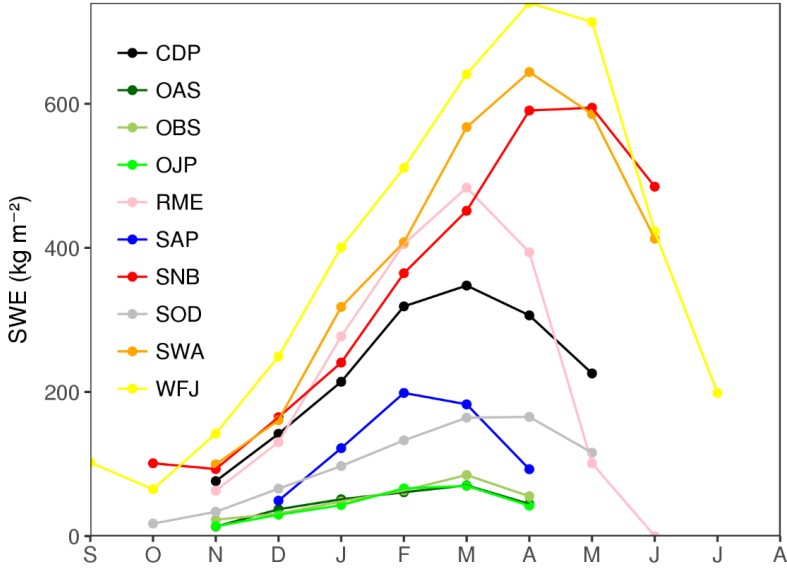


*Figure 5: Monthly climatological averages of manual snow water equivalent measurements.*

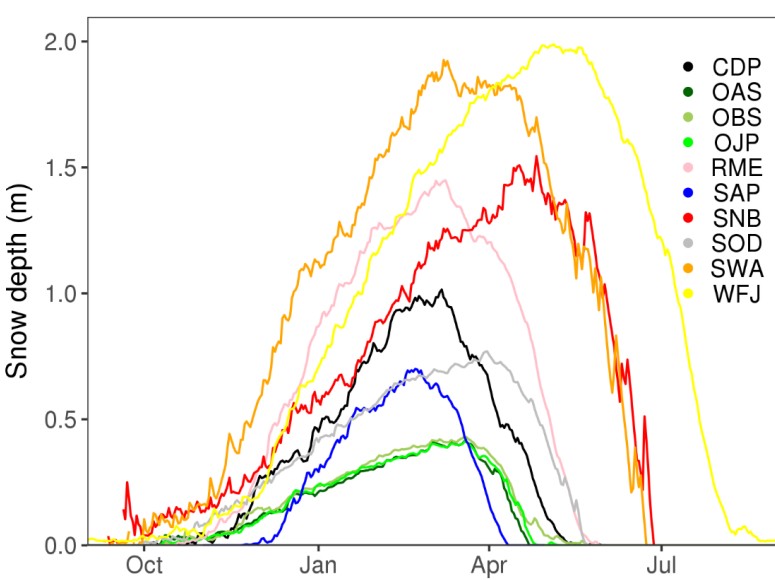

*Figure 6: Daily climatological averages of snow depth measurements at all sites.*



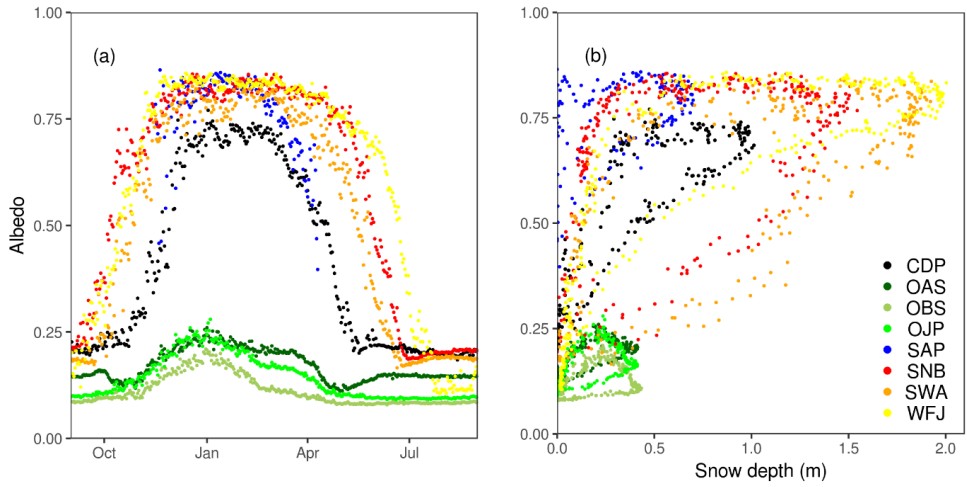


Figure 7: Daily averaged albedo over time (a) and as a function of snow depth (b) at all sites except RME and SOD.



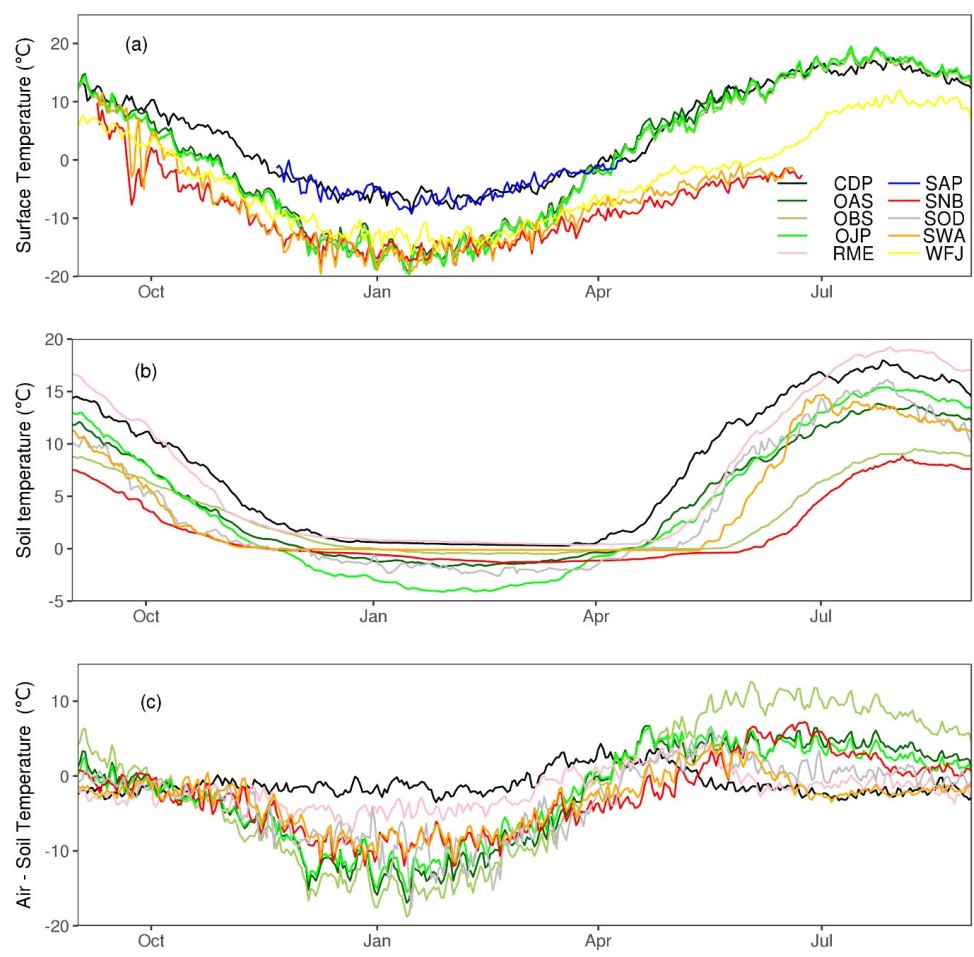

*Figure 8: Daily climatological averages of surface temperature (a), soil temperature (b), and differences between air and soil temperatures (c). Soil temperatures are shown at 30 cm depth at RME and at 10 cm depth at all other sites.*





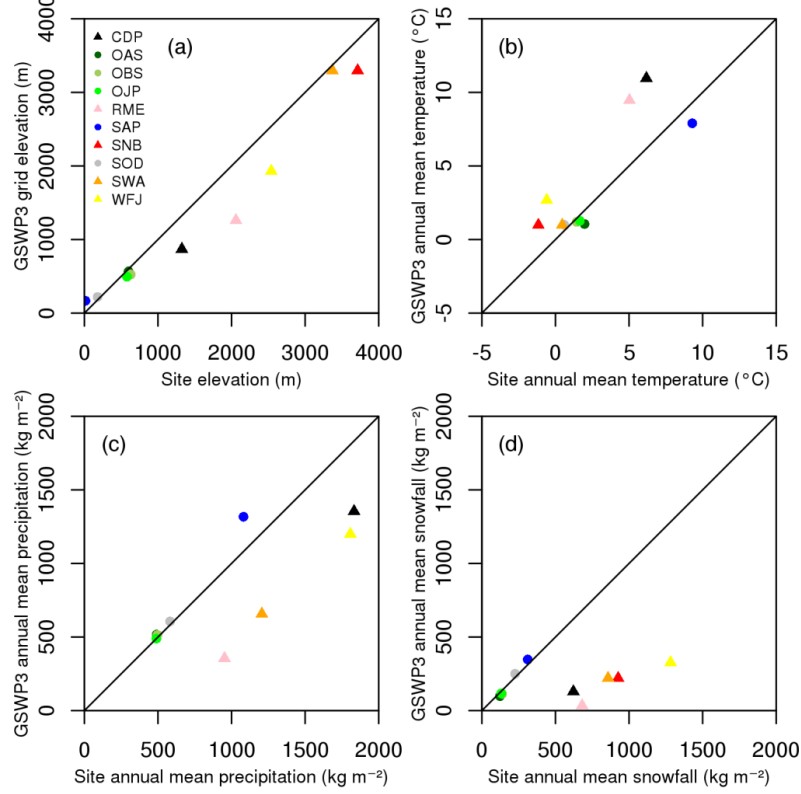

*Figure 9: Comparisons between elevations (a), temperatures (b), total precipitation (c) and snowfall*
*(d) at ESM-SnowMIP reference sites and corresponding GSWP3 grid cells. Triangles identify mountain*
*sites.*





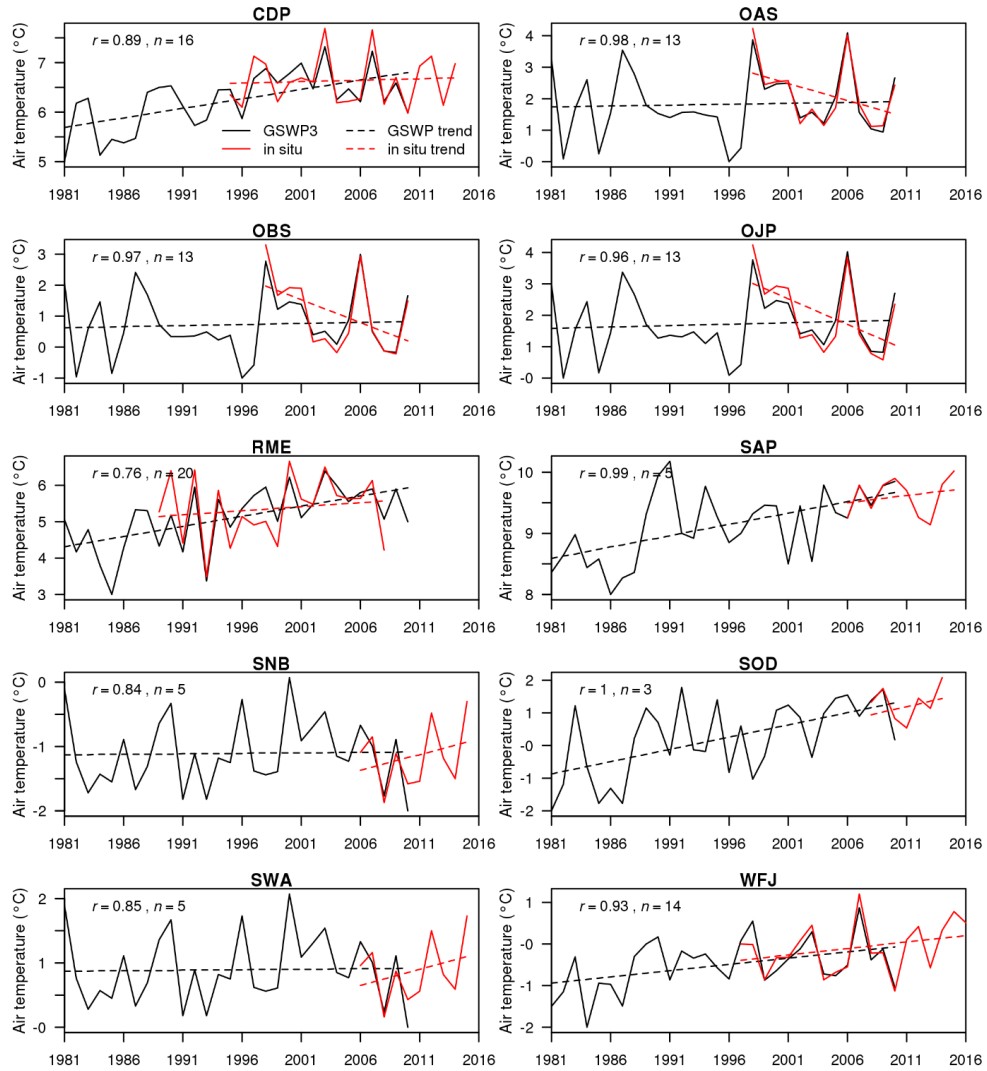

*Figure 10: Annual mean temperatures and fitted trends for years starting on 1 October at reference sites from GSWP3 and in situ data. Numbers show correlation (r) between GSWP3 and*
*in situ air temperature for the n complete years of overlap.*



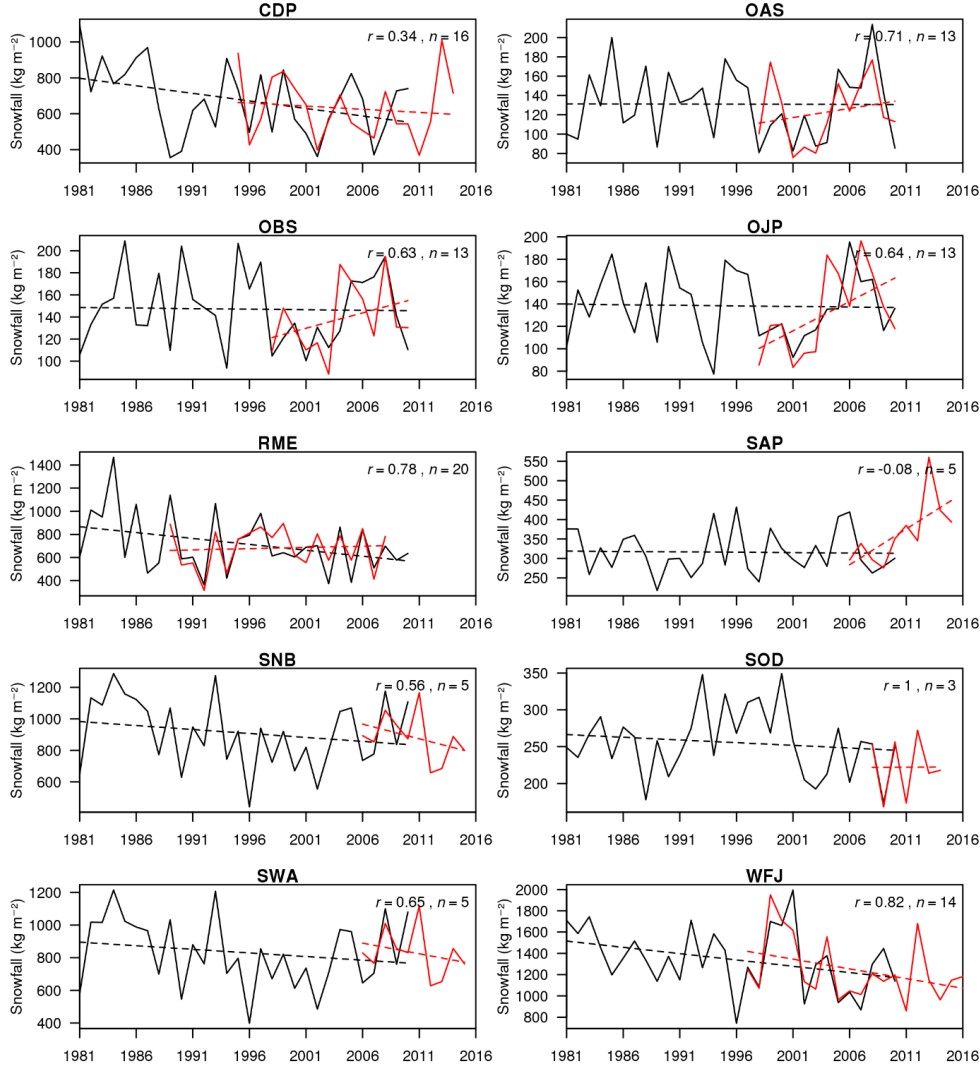

*Figure 11: Annual snowfall and fitted trends for years starting on 1 October at reference sites from GSWP3 and in situ data. Numbers show correlation (r) between GSWP3 and in situ snowfall for the n complete years of overlap. The legend is as in Figure 10.*
