# Peer review of "Meteorological and evaluation datasets for snow modelling at ten reference sites: description of in situ and bias-corrected reanalysis data"

_Earth System Science Data, 2019_

## Referee Comment (RC1) · Anonymous Referee #1 · 15 Apr 2019

The paper entitled "Meteorological and evaluation datasets for snow modelling at ten reference sites: description of in situ and bias-corrected reanalysis data" is an overall description of ten datasets that have been somewhat standardized to ease their use as benchmarks for model developments. The work undertaken by the authors is of crucial importance to the scientific community and this paper will certainly help future users to better understand how to use these datasets. The discussion is particularly well written and enlightening. Section 2 needs a bit of work in terms of flow of ideas, as it appears to have been written by multiple authors with abrupt (and sometimes

confusing) transitions between description elements.

I therefore recommend this paper for minor revisions.

Major comments:

Section 2 seems to be written by multiple authors and is difficult to read due to abrupt transitions. As an example, the sentence starting on line 128 should be the beginning of a new paragraph, as the reader may wonder if that sentence refers to Sodankyla only.

Line 202: add a note that the peculiar behavior seen in the SAP site will be discussed later on, or better yet, give the reader a quick explanation.

Lines 207-210: I don't think that differences between annual snowfall and peak SWE should be termed discrepancies. A number of processes can lead to these differences, such as sublimation, rain-on-snow events, interception, melting events, etc., all of which are natural and not erroneous as the word "discrepancies" implies. I certainly would not expect a perfect match between annual snowfall and peak SWE.

Section 2.1.5: reading the text, it is not clear whether the datasets provide relative humidity, specific humidity, or both (which would be best, as it would allow users to choose). Please clarify.

Line 339: What type of bias corrections do you refer to? Is it what is described in the paragraph starting on line 358? If so, please mention it, otherwise please elaborate.

Line 293-294: Is the rejection applied to all sites? Please clarify.

Minor comments:

Line 190: underestimated

Line 274: Please add the table number.

Line 380: Please add table number.
Table 3: Third column, replace Snow by snow. Last column, what is the number 670 referring to?

---

## Referee Comment (RC2) · Anonymous Referee #2 · 26 Apr 2019

This paper deals with the data sets of ten observation sites under different climate conditions, which includs detailed quality information of each sites. It is clear to see that a lot of hard work has been put into the study. I do not doubt this paper should widely contribute to the scientific community for not only comparison and improvement of models but also understanding the response of Cryosphere to climate change. The paper has enough scientific quality, however, some of parts need to be improved because of ambiguous descriptions. Thus, several points as indicated below need to be addressed by authors to improve the quality of the article. For this reason, I concluded

that the status of this paper is minor revisions.

Major comments:

Section 2 is not designed well because the information of each site appears sporadically and description style is not unified. Moreover, the definitions of several terms are ambiguous. Therefore, I recommend to rewrite this section for better understanding.

L139: "Monthly averages" How long period for average? Averaged period depends on the site? Please add the information.

L149 : What is the definition of air temperature in Fig. 2? Is it daily or hourly? Please add the definition.

L170-L172. It should be helpful for understanding if the authors add a figure showing some evidences for discussing in this paragraph.

L217: what is dimension of 1/10 ?

Fig. 7 What is the difference between "daily averaged over time" and "daily climatological averages" ? Please clear the definition.

Line 328-330: It is difficult to understand this sentence. Are the data of soil temperature at OJP and SOD discussed in this sentence? Please clear.

L352: How to calculate snowfall amount in GSWP3? The short introduction of this information is useful for better understanding of not only Fig. 9 but also Fig. 3.

LL371: The result of Sapporo at Fig. 3 also has strange behaviors: Solid traction of precipitation keeps large value (around 0.1) at the area of the higher than 5 degree Celsius. I recommend to check the data of GSWP3 at Sapporo.

Minor comment:

Several numbers of tables or figure in the text are missing (e. g. L 147, L 274, L380). Please carefully check them and add the numbers.

---

## Author Comment (AC1) · 20 May 2019

**Referee #1:**

Major comments:

Section 2 seems to be written by multiple authors and is difficult to read due to abrupt transitions. As an example, the sentence starting on line 128 should be the beginning of a new paragraph, as the reader may wonder if that sentence refers to Sodankylä only

Thank you for your suggestion. Acknowledging that comprehensive descriptions of the forcing meteorological data were already provided in individual data papers, we reorganised Section 2.1. focusing on presenting the data in a way which we believe will be more helpful to users in identifying specificities and differences about the sites as well as potential sources of uncertainty in the datasets. Individual sections for the forcing variables were therefore removed and replaced with: 2.1.1 Differences and similarities between sites, 2.1.2 Issues with instrumentation, 2.1.3 Modelling and modification to in situ data. Note that, although the text was reorganised, the information provided has not. Section 2.2. Evaluation Data still has individual sections about the evaluation data as we believe it provides the most comprehensive format for users. However, some of the text was edited to improve readability.

Line 202: add a note that the peculiar behavior seen in the SAP site will be discussed later on, or better yet, give the reader a quick explanation.

What is discussed later relates to the GSWP3 data, not to in situ data, therefore it wasn't deemed appropriate to add anything in this section.

Lines 207-210: I don't think that differences between annual snowfall and peak SWE should be termed discrepancies. A number of processes can lead to these differences, such as sublimation, rain-on-snow events, interception, melting events, etc., all of which are natural and not erroneous as the word "discrepancies" implies. I certainly would not expect a perfect match between annual snowfall and peak SWE.

Good point. References of "discrepancies" were removed altogether.

Section 2.1.5: reading the text, it is not clear whether the datasets provide relative humidity, specific humidity, or both (which would be best, as it would allow users to choose). Please clarify.

Clarified in the text (line 197) which is provided and why specific humidity only is available in the dataset.

Line 339: What type of bias corrections do you refer to? Is it what is described in the paragraph starting on line 358? If so, please mention it, otherwise please elaborate.

It isn't the same bias correction mentioned on line 358. This was clarified in the text.

Line 293-294: Is the rejection applied to all sites? Please clarify.

Yes. Clarified in the text.

Minor comments:

Line 190: underestimated. Done

Line 274: Please add the table number. Done

Line 380: Please add table number. Done

**Referee #2:**

Major comments: Section 2 is not designed well because the information of each site appears sporadically and description style is not unified. Moreover, the definitions of several terms are ambiguous. Therefore, I recommend to rewrite this section for better understanding.

Thank you for your suggestion. Acknowledging that comprehensive descriptions of the forcing meteorological data were already provided in individual data papers, we reorganised Section 2.1. focusing on presenting the data in a way which we believe will be more helpful to users in identifying specificities and differences about the sites as well as potential sources of uncertainty in the datasets. Individual sections for the forcing variables were therefore removed and replaced with: 2.1.1 Differences and similarities between sites, 2.1.2 Issues with instrumentation, 2.1.3 Modelling and modification to in situ data. Note that, although the text was reorganised, the information provided has not. Section 2.2. Evaluation Data still has individual sections about the evaluation data as we believe it provides the most comprehensive format for users. However, some of the text was edited to improve readability.

L139: "Monthly averages" How long period for average? Averaged period depends on the site? Please add the information.

Averaged periods do depend on the length of the data sets at individual sites. This was clarified in the text (lines 122-124).

L149 : What is the definition of air temperature in Fig. 2? Is it daily or hourly? Please add the definition.

Hourly. Added.

L170-L172. It should be helpful for understanding if the authors add a figure showing some evidences for discussing in this paragraph.

This was shown in Figure 1, although the figure wasn't explicitly referenced at this point in the text. However, following the reorganisation of Section 2.1, this information was not deemed to be of interest to users and was removed altogether.

L217: what is dimension of 1/10 ?

Like 2/3 earlier in the sentence, it is dimensionless.

Fig. 7 What is the difference between "daily averaged over time" and "daily climatological averages"? Please clear the definition.

The caption should have been "Daily climatological averages". This was corrected.

Line 328-330: It is difficult to understand this sentence. Are the data of soil temperature at OJP and SOD discussed in this sentence? Please clear.

The sentence was re-written and the references to OJP and SOD clarified (lines 332-334).

L352: How to calculate snowfall amount in GSWP3? The short introduction of this information is useful for better understanding of not only Fig. 9 but also Fig. 3.

A short introduction to the generation of GSWP3 data is in (lines 341-347). We have added that precipitation is bias-corrected using observations, but the full details of how snowfall is calculated and bias-corrected can be found in Weedon et al. (2011) and Kim et al. (2017), now both cited in the paper.

LL371: The result of Sapporo at Fig. 3 also has strange behaviors: Solid traction of precipitation keeps large value (around 0.1) at the area of the higher than 5 degree Celsius. I recommend to check the data of GSWP3 at Sapporo.

We have checked the GSWP3 data and they do show these unusual patterns. The developers of the GSWP3 data (the PI, Dr Kim, is one of the co-authors) are aware of this issue and will investigate for future releases of the product.

Minor comment: Several numbers of tables or figure in the text are missing (e.g. L 147, L 274, L380).Please carefully check them and add the numbers. Done.